# Enteric Methane Emissions and Animal Performance in Dairy and Beef Cattle Production: Strategies, Opportunities, and Impact of Reducing Emissions

**DOI:** 10.3390/ani12080948

**Published:** 2022-04-07

**Authors:** Byeng-Ryel Min, Seul Lee, Hyunjung Jung, Daniel N. Miller, Rui Chen

**Affiliations:** 1College of Agriculture, Environment and Nutrition Sciences, Tuskegee University, Tuskegee, AL 36088, USA; rchen@tuskegee.edu; 2Animal Nutrition & Physiology Division, National Institute of Animal Science, Rural Development Administration, Wanju-gun 55365, Jeollabuk-do, Korea; tabababy@korea.kr (S.L.); hyjjung@korea.kr (H.J.); 3Agroecosystem Management Research Unit, USDA/ARS, 354 Filly Hall, Lincoln, NE 68583, USA; dan.miller@usda.gov

**Keywords:** beef cattle, dairy cattle, methanogenesis, rumen, average daily gain, milk production

## Abstract

**Simple Summary:**

Numerous enteric methane (CH_4_) mitigation opportunities exist to reduce enteric CH_4_ and other greenhouse gas emissions per unit of product from ruminants. Research over the past century in genetics, animal health, microbiology, nutrition, and physiology has led to improvements in dairy and beef cattle production. The objectives of this review are to evaluate options that have been demonstrated to mitigate enteric CH_4_ emissions per unit of products (energy-corrected milk, milk yield, average daily gain, dry matter intake, and gross energy intake) from dairy and beef cattle on a quantitative basis and in a sustained manner, and to integrate approaches in feeding, rumen fermentation profiles, and rumen microbiota changes to emphasize the understanding of these relationships between enteric CH_4_ emissions and animal productivities.

**Abstract:**

Enteric methane (CH_4_) emissions produced by microbial fermentation in the rumen resulting in the emission of greenhouse gases (GHG) into the atmosphere. The GHG emissions reduction from the livestock industry can be attained by increasing production efficiency and improving feed efficiency, by lowering the emission intensity of production, or by combining the two. In this work, information was compiled from peer-reviewed studies to analyze CH_4_ emissions calculated per unit of milk production, energy-corrected milk (ECM), average daily gain (ADG), dry matter intake (DMI), and gross energy intake (GEI), and related emissions to rumen fermentation profiles (volatile fatty acids [VFA], hydrogen [H_2_]) and microflora activities in the rumen of beef and dairy cattle. For dairy cattle, there was a positive correlation (*p* < 0.001) between CH_4_ emissions and DMI (*R*^2^ = 0.44), milk production (*R*^2^ = 0.37; *p* < 0.001), ECM (*R*^2^ = 0.46), GEI (*R*^2^ = 0.50), and acetate/propionate (A/P) ratio (*R*^2^ = 0.45). For beef cattle, CH_4_ emissions were positively correlated (*p* < 0.05–0.001) with DMI (*R*^2^ = 0.37) and GEI (*R^2^* = 0.74). Additionally, the ADG (*R*^2^ = 0.19; *p* < 0.01) and A/P ratio (*R*^2^ = 0.15; *p* < 0.05) were significantly associated with CH_4_ emission in beef steers. This information may lead to cost-effective methods to reduce enteric CH_4_ production from cattle. We conclude that enteric CH_4_ emissions per unit of ECM, GEI, and ADG, as well as rumen fermentation profiles, show great potential for estimating enteric CH_4_ emissions.

## 1. Introduction

Ruminant animal production is dependent on the anaerobic microbial ecosystem (including bacteria, archaea, protozoa, and fungi) to ferment and transform human indigestible forages into high-grade dairy and meat products for human consumption. Ruminant animals, however, are major emitters of enteric methane (CH_4_) due to the microbial breakdown of carbohydrates in the rumen [1,2], representing an unproductive loss of dietary energy [3]. The rumen microbial fermentation process, also referred to as enteric fermentation, produces various gases, including carbon dioxide (CO_2_) and CH_4_, as by-products, exhaled or eructated by the ruminant (Table 1). The eructation of gases via belching is important in bloat prevention and a primary route for CH_4_ emission to the atmosphere [4]. Estimates of the gas production rate in cattle range from less than 0.2 L/min in the fasted animal to 2.0 L/min following feeding [5]. Generally, lower feed quality and higher feed intake lead to higher CH_4_ emissions [1]. Although feed intake is positively correlated with animal size, growth rate, level of activity, and production (e.g., milk production, wool growth, pregnancy, or work [6]), it also varies among animal types and management practices for individual animal types (e.g., cattle in feedlots or grazing on grassland). From an energy perspective, enteric CH_4_ emissions associated with rumen fermentation activities result in the loss of 6–12% of gross energy intake (GEI), or 8–14% of the digestible energy intake (DEI) of ruminants [3,7,8], which could, in principle, otherwise be available for animal growth or milk production. Reducing enteric CH_4_ emissions from cattle would benefit the environment and improve meat and milk production’s efficiency and economic profitability.

Livestock production systems face challenges posed by increasing food demand and environmental issues. When animal productivity is improved through nutrition, feeding management, reproduction, or genetics, CH_4_ production per unit of meat or milk is reduced [9]. Beauchemin and McGinn [10] estimated that a 20% reduction in CH_4_ production could allow growing cattle to gain an additional 75 g/d of body weight and 1 L/d more milk yield (MY) from dairy cows. Although total CH_4_ emissions in cattle fed full mixed rations (TMR) increase with increasing concentrate feed levels [11,12,13,14], emissions per unit of milk produced [15], or emissions per kg of average daily gain (ADG [16]) generally decrease. However, much less evidence exists concerning the effect of dry matter intake (DMI), feed efficiency, rumen fermentation profiles, rumen microbiome changes, and enteric CH_4_ emissions per unit of ADG or MY (CH_4_ intensity; g CH_4_/kg of MY) from dairy and beef cattle, respectively [16,17,18].

Several reviews of enteric CH_4_ production from cattle have been published [1,16,19,20,21]. Unlike this review, they all focus more on mitigation options than understanding relationships among dietary and rumen properties that lead to CH_4_ production associated with enteric CH_4_ emissions factors (*Y*m; % GEI) and CH_4_ emissions intensity (product yield [16,20]). This review aims to explain how enteric CH_4_ emissions are associated with DMI, GEI, ADG, MY, energy-corrected milk (ECM), rumen fermentation rate, and ruminal microbiota changes in dairy and beef cattle fed forage- and grain-based diets. The improved understanding of these relationships between enteric CH_4_ emissions and animal productivities may provide insights into cost-effective means to reduce enteric CH_4_ production.

## 2. Interrelationships between Methane (CH_4_) Production, Dry Matter Intake (DMI), and Gross Energy Intake (GEI)

In this analysis, a database of several studies examining the effects of mitigation strategies on enteric CH_4_ emissions per unit of milk production, ADG, DMI, and GEI in dairy cows (Table 2 and Table 3) and beef cattle (Table 4 and Table 5) was created with enteric CH_4_ emissions per unit of ECM (CH_4_/kg of ECM) (Table 2 and Table 6) and rumen fermentation parameters (Table 7) are also evaluated. Statistical analyses of the dataset [16,20] included calculations of slopes, correlation coefficients, and regression coefficients using the Proc Corr. procedure (SAS Institute Inc., Cary, NC, USA). A simple regression analysis using Proc Reg in SAS (SAS Institute Inc., Cary, NC, USA) was conducted to evaluate how DMI, GEI, milk production, ADG, and rumen fermentation profiles were related to CH_4_ emissions from cattle (Figure 1, Figure 2, Figure 3, Figure 4, Figure 5 and Figure 6). An ordinary least squares regression (OLS) was also used to estimate the impacts of animal performance on the enteric CH_4_ emission in dairy and beef cattle, respectively (Table 3, Table 5, Table 6, Table 7 and Table 8), used in Equation (1):(1)$$Y_{i}=\beta_{0}+\beta_{1}X_{i}+\varepsilon_{i}$$
where *Y_i_* denotes CH_4_ production (enteric CH_4_ emissions) per unit of output from dairy/beef cattle*_i_*, *X_i_* is the animal performance of cattle*_i_* (such as dry matter intake (DMI*_i_*), gross energy intake (GEI*_i_*), milk production*_i_*, ADG*_i_*, proipionate*_i_*, A/P*_i_*). The impact(s) of animal performance on enteric CH_4_ emissions is/are denoted by $\beta_{i}$. In each analysis, a test the null hypothesis that $\beta_{1}$ is zero was evaluated. When the regression analysis was conducted using Table 3 and Table 4, the null hypothesis that animal performance had no impact on enteric CH_4_ emissions was rejected, as shown in Table 3, Table 4, Table 5, Table 6, Table 7 and Table 8 and Figure 1, Figure 2, Figure 3, Figure 4 and Figure 5. That is to say, CH_4_ production (g/d) was significantly correlated with the animal performance- DMI*_i_*, GEI*_i_*, milk production*_i_*, ADG*_i_*, propionate*_i_*, or A/P*_i_*.

In temperate regions, our estimates of ${\mathrm{DMI}}_{i}$ have an impact on CH_4_ emissions (18.53 and 18.93 g of CH_4_/kg DMI for dairy and beef cattle, respectively; Table 2) and were similar to the range of 19.6 to 21.5 g/kg DMI found in previously published studies [73,74,75,76]. This is consistent with both dairy cattle (fed temperate forages) and beef cattle (fed temperate and tropical forages) studies and reported that the relationships between CH_4_ production and DMI were very similar (CH_4_ production (g/day) = 20.7 ± 0.28 × DMI (kg/d); *R*^2^ = 0.92, *p* < 0.001) for all three production categories [73]. However, individual determinations of enteric CH_4_ carried out in respiration chambers found that the average CH_4_ production for cattle (e.g., Brahman steers) fed tropical grasses ranged from 19.3 to 34.1 g CH_4_/kg DMI [77], indicating that tropical (C4) grasses contribute to enteric CH_4_ emissions to a greater extent than temperate (C3) grasses [78]. This is probably due to the difference in dietary composition between typical diets in temperate grasses (high-quality grasses) and tropical grasses (low-quality grasses), and the digestibility of these diets. Previously published studies showed variance in CH_4_ production values from beef cattle, due to different CH_4_-measurement methods, age, feed type, cattle breeds, day-to-day variations, individual physiological stage, and metabolic BW [3,6,20,36,73,79,80,81,82]. The model of Chamley et al. [73] also reported that these factors might mutually present an error of ~13.4% in predicting CH_4_ emissions for individual animals. In the present study, measurements in the above dataset were from lactating Holstein–Friesian, Jersey, and cannulated dairy cows with a high DMI and high CH_4_ production. The beef dataset consisted of growing/finishing steers or non-lactating heifers with lower BW and DMI and low CH_4_ production. Data included CH_4_ measurements from indoor respiration chambers (RC), using the sulfur-hexafluoride (SF_6_) method, and the GreenFeed method (GF; C-Lock Inc., Rapid City, SD, USA), which may account for some of the variances in the dataset. It should be noted that Hammond et al. [39,83] used RC for the silage study, while the SF_6_ technique was used for the grazing study. Recently, Min et al. [82] indicated that the three different CH_4_-measurement methods (RC, SF_6_, and GF) might be highly variable in the relationship between daily CH_4_ production and DMI (g/kg DMI). Based on Hammond et al. [68,84], the average estimate of CH_4_ production (g/d) varied among the three measurement techniques (RC, SF_6_, and GF). 

When the regression analysis was conducted using the data in Table 2 and Table 4, CH_4_ productions (g/d) were significantly correlated with DMI_*i*_, and GEI_*i*_ in dairy and beef cattle (Table 2, Table 3, Table 4 and Table 5 and Figure 1a–d), respectively. In agreement with others, animal feed intake, either as GEI or DMI, had a strong linear relationship with CH_4_ production: models based on these variables were of comparable accuracy with negligible bias [80,85,86]. In the present analysis, total CH_4_ production (g/d) increased with increasing DMI (Figure 1a,b) and GEI (Figure 1c,d) in dairy and beef cattle, simply because there was more feed available for rumen fermentation. Johnson and Johnson [3] reported that, for each kg of increase in DMI, there was, on average, a 1.6% decrease of feed gross energy (GE) lost as CH_4_. One study found a 2.1% reduction in the CH_4_ conversion factor (*Y*_m_; the proportion of the GEI converted to enteric CH_4_ energy) per kg of DMI increase from dairy cows [87]. Typical ruminant diets contain about 18.4 MJ of GE per kg of DM, and CH_4_ has an energy content of 55.65 MJ/kg [88]. The IPCC [89] recommends *Y*_m_ ranges of 3.0 ± 1.0% GEI lost as CH_4_ for feedlot cattle and 6.5 ± 1.0% GEI lost as CH_4_ for dairy and other well-fed cattle consuming temperate-climate feed types [89]. However, the *Y*_m_ does not consider other relevant animal or dietary characteristics that impact CH_4_ emissions, such as digestibility, rumen fermentation characteristics, nutrient profiles, microbial community structure, diet composition, or cattle management.

The annual global CH_4_ emission from dairy cows is approximately 18.9 Tg [90], representing a loss of 5.5–6.5% of dietary GEI [91]. However, CH_4_, as a proportion of DMI or GEI (CH_4_/kg of GEI), usually decreases as DMI increases above maintenance [69,92,93], and is related to decreased DM digestibility at higher DMI [1]. It has been reported that CH_4_ production decreases with increasing levels of dietary concentrate fed [94] and can be as low as 3% of GEI [3] for diets with a high proportion (>60%) of concentrate. Metabolizable energy intake (MEI), neutral detergent fiber (NDF), acid detergent fiber (ADF), ether extract, lignin, and forage proportion need to be considered in the development of models to predict CH_4_ emissions [95]. Although the information on milk production would be relevant to assess the impact of animal performance on CH_4_ estimates, data on milk production, ADG, rumen fermentation characteristics, and microbiome changes in CH_4_ studies were insufficient.

## 3. Enteric Methane (CH_4_) Emissions, Milk Production, and Average Daily Gain (ADG) in Dairy and Beef Cattle

Numerous studies reported that a close relationship exists between DMI and milk production of dairy cows [96,97,98,99,100], but limited information is available to calculate the relationships between milk production and CH_4_ emissions in dairy cattle or ADG and CH_4_ emissions in beef cattle. It has been reported that a linear relationship (*R*^2^ = 0.47) exists between DMI and milk production [101,102]. The current analysis confirms a positive relationship (*p* < 0.01; Figure 2a) between DMI and milk production (Table 5) in dairy cattle (y = 1.31x + 1.34 ± 2.70; *R*^2^ = 0.34; *p* < 0.001). We found that, as DMI increased by 1.0 kg/d, there was a 1.31 kg/d increase in milk production in dairy cattle (Figure 2a). This agrees with Trupa et al. [103], who proposed that, for every 2 kg of milk production, a cow consumes at least 1 kg of DMI (legume hay + concentrate). It has been documented that pasture DMI generally decreases when grazing cows are offered concentrate supplements, whereas total DMI and milk yield increase with concentrate feeding [104]. This analysis confirmed this positive relationship (Table 5; Figure 2a). Min et al. [105] reported that milk production increased by 1.7 and 0.9 kg for each additional kg of concentrate fed per day during the first and second years of lactation by dairy goats, respectively. The same authors reported that improved nutrition leads to an increase in daily milk yield (22%), peak yield (17%), time of peak yield (14 d), and persistency (8%; as the ability of a cow to continue milk production at a high level after the peak yield), compared with control treatment.

For our dataset, we found a positive relationship (Table 6; Figure 2b) between DMI and ADG (kg/d) in beef cattle (y = 0.09x + 2.44 ± 0.98; *R*^2^ = 0.50; *p* < 0.01), whereas DMI increased by 1.0 kg/d, and there were a 0.09 kg/d increase in ADG in beef cattle fed mixed (grazing + feedlot) diets (Figure 2b). Other studies reported that each 1 kg increase in DMI increases ADG by 0.08–0.09 kg/d (silage-based diet) and 0.14–0.16 kg/d (grain-based diet) in finishing cattle [59,60,106]. Along with DMI, intake of dietary energy and protein, or individual carbohydrate and protein contents, environmental stress, ration palatability, and feed processing may be important factors affecting milk and meat production, and require further analyses in the future [103,107]. The dietary energy associated with animal maintenance is about 70–75% in beef cattle and 50% in dairy cattle [105]. The remaining nutritional energy is used to produce meat, milk, or gestation. Thus, as productivity increases, CH_4_ emissions also increase (Figure 3a,b), but CH_4_ emissions per unit of product decrease [106]. 

When the regression analysis was conducted on our dataset (Table 3 and Table 4), milk production was associated (*p* < 0.001) with CH_4_ production (Figure 3a; y = 9.82x + 142. 69 ± 33.55); *R*^2^ = 0.37) in dairy cattle (Table 6). The ADG (kg/d) was also associated (*p* < 0.01) with CH_4_ emission (Figure 3b; y = 117.33x + 38.34 ± 53.7); *R*^2^ = 0.19) in beef steers (Table 6). Despite significance from the combined estimated slope (Figure 3a), the relationship between milk production and CH_4_ production in a grain-based diet (Figure 3c) is not significant (*p* = 0.12). However, there was a significant difference (*p* < 0.01) in CH_4_ emissions per kg ADG in beef cattle (*R*^2^ = 0.38–0.40) fed grain-based (Figure 3e) and forage-based (Figure 3f) diets. This dataset took measurements on lactating Holstein–Friesian, Jersey, and cannulated dairy cows on high-quality dairy rations with some silage (e.g., corn, wheat, or grass silages) supplementation or high-quality grazing forage (e.g., alfalfa). These animals were found to have similar CH_4_ production between high-forage and low-forage diets. In contrast, measurements in the beef dataset were from growing/finishing steers or non-lactating heifers with two different energy content diets (e.g., high forage- and high grain-based diets) that had significantly different CH_4_ production between forage-based and grain-based diets. Adding grain to the feed ration increases the starch content. It reduces the amount of crude fiber, reducing rumen pH and promoting propionate production in the rumen while reducing the CH_4_ yield [103]. McGeough et al. [60,107] reported in their study that CH_4_ emissions from beef cattle increased from 15.3 g/kg DMI for ad libitum concentrates to 25.9–30.1 g/kg DMI for whole crop wheat silage diets using the SF_6_ technique. These data are comparable to those documented in the current study. Likewise, McGeough et al. [60,107] reported that CH_4_ emissions increased from 22.1 g/kg DMI for the ad libitum grain-based diet to 26.2–29.4 g/kg DMI for diets based on corn silage from crops at various growth stages at harvest (supplemented with concentrates at 0.23 to 0.25 g/kg DM of the diet). Therefore, diet quality and ingredients have substantial effects on CH_4_ production: if the feed quality is poor (e.g., high forage), the production of CH_4_ is high (Figure 3d,f). This is the primary cause of the loss of cow energy and, if it could be avoided, it would be critical to attaining increases in the ADG or milk production. However, improving productivity with the use of high-grain diets must be evaluated in terms of the cost of feed production and the use of fertilizers and machinery, which will increase fossil fuel use and increase N_2_O emissions.

Research over the past century in dietary interventions, animal genetics, modified rumen microbial community structure, nutrition, and physiology has led to improvements in dairy production. Intensively managed dairy farms have GHG emissions as low as 1 kg of CO_2_ equivalents (CO_2_e)/kg of ECM, compared with >7 kg of CO_2_eq/kg of ECM in less extensively managed farms [1]. High-quality grain-based diets deliver more energy for animal production as a proportion of the GEI or DMI (kg/d), and dilute the costs of maintenance more than low-quality forage-based diets or grazing, resulting in lower CH_4_ g/kg ECM (Table 8; Figure 4), consistent with Knapp et al. [1]. Accordingly, we found that CH_4_ g/d decreased (*p* < 0.001; *R*^2^ = 0.46) with increasing ECM, g/kg in dairy cattle (Figure 4). As a result, the enteric CH_4_ emissions per unit of ECM (CH_4_/ECM) are useful measurements in biology, nutrition, environmental quality, and economics [1]. These data indicated that altering the forage quality and forage-to-concentrate ratio can affect enteric CH_4_ emissions. Forage feeds are high in NDF, ADF, and lignin, which are more difficult to digest than concentrates [60]. The slower digestion of a forage-based diet results in higher acetate formation in the rumen, and produces more CH_4_ than the faster digestion of a grain-based diet (Figure 4). Grain-based diets are high in starch and soluble carbohydrates and are more digestible than fibrous forage-based diets [60]. It has been reported that a higher forage-to-concentrate ratio in the diets increases enteric CH_4_ emissions and may decrease milk production depending upon the quality (digestibility) of the forage [1]. Aguerre et al. [14] found that enteric CH_4_ emissions increased by 20% when increasing the forage-to-concentrate ratio from 47:53 to 68:32. However, grain-based diets can be more expensive, decrease milk fat content, and result in metabolic disorders [107]. 

Alterations in milk pricing, from systems based on butterfat content to systems based on protein or other milk components, have been recommended to reduce CH_4_ emissions [106]. The fat content of milk accounts for about 9253 calories per gram of fat or 750 calories per 1 kg of 4% milk of the energy content of milk, and therefore reducing milk fat content will decrease the need for feed energy [108], which, sequentially, will reduce enteric CH_4_ emissions. A change in milk pricing based on solid-non-fat has been projected to reduce CH_4_ emissions from U.S. milk cows by 15% [106]. With the application of low-fat milk increasing, pricing based on milk protein will increase producers to adapt feeding systems to include highly digestible protein feeds, which will increase productivity and reduce CH_4_ emissions. However, high protein ingredients are expensive in dairy rations, and excessive nitrogen (N) may be excreted in urine and feces. The impact on the environment as well as dietary feed accounts associated with such an approach must be assessed in terms of the overall profits that can be attained. 

## 4. Enteric Methane Emissions and Rumen Fermentation Profiles

To further explore the effect of energy sources, as measured by volatile fatty acids (VFA; Figure 5a–d) and acetate/propionate (A/P) ratio (Figure 5e,f) on CH_4_ emissions, these values were regressed against CH_4_ in dairy and beef cattle in the study dataset (Table 7). We found that there was a negative correlation between propionate concentration and CH_4_ emissions in dairy (*R*^2^ = 0.21; *p* < 0.001; Figure 5a) and beef cattle (*R*^2^ = 0.21; *p* < 0.02; Figure 5b), and a positive correlation between acetate and CH_4_ productions (more acetate, more CH_4_ in the rumen) in dairy (*R*^2^ = 0.28; *p* < 0.001; Figure 5c) and beef cattle (*R*^2^ = 0.10; *p* = 0.10; Figure 5d), which is similar to the A/P ratio (*R*^2^ = 0.45–0.15; *p* < 0.001–0.05; Figure 5e,f) and CH_4_ emissions in dairy and beef cattle, respectively. Acetate is the most important intermediate substrate of CH_4_ production (acetoclastic methanogenesis or syntrophic acetate oxidation coupled with hydrogenotrophic methanogenesis) during anaerobic digestion and the biogas process [109]. Aceticlastic methanogenesis is carried out by *Methanosarcinaceae* spp. and *Methanosaetaceae* spp., while syntrophic acetate oxidation is performed by methanogens (mediated by *Methanobacteriales* spp. and/or *Methanomicrobiales* spp.) and acetate-oxidizing bacteria, including *Clostridium ultunense*, *Syntrophaceticus schinkii*, *Tepidanaerobacter acetatoxydan,* and other thermophilic bacterial species [110,111,112,113,114]. Likewise, Kittelmann et al. [115] proposed that proportionally more propionate was present in one of the low CH_4_ emitting cattle types in that study. Intrinsically, a dietary element or intervention that initiates a shift in support of propionate production will yield a reduction in CH_4_ production per unit of feed fermented. In contrast, the opposite is true for acetate and butyrate [115]. Danielsson et al. [116] reported that the ruminal fermentation pattern of VFA showed that the proportion of propionate was higher in cluster L cows (low-CH_4_ production), while the proportion of butyrate was higher in cluster H cows (high-CH_4_ production). As a result, propionate fermentation is the most energy-efficient fermentation process due to energy assimilation from H_2_ and propionate being the main precursor of gluconeogenesis in animals [117,118]. This phenomenon at least partially explains the relationship between propionate concentration, the A/P ratio, and CH_4_ production observed in this study (Figure 5e,f). Rumen fermentation that leads to propionate synthesis results in less H_2_ being available for CH_4_ production [115,119], which is primarily formed using H_2_ by methanogenic archaea (CO_2_ + 4H_2_ – CH_4_ +2H_2_O [120]).

Weimer et al. [121] observed that the ruminal total VFA concentration and propionate proportion were higher in highly efficient cows than in low-efficiency cows. The primary energy sources for dairy and beef cattle are carbohydrates. Rumen microbes ferment these energy sources in the rumen to produce VFA (up to 200 mM) and various gases (Table 1), which are used by ruminants as the energy source for milk and meat production, resulting in up to 75% of the cow’s metabolizable energy requirement [117,118]. It is reported that, as ruminal VFA production moves towards more propionate at the cost of acetate (e.g., a lower A/P), more ADG is achieved, and presumably more energy is utilized for animal growth [115]. When glucose is metabolized into acetate, propionate, or butyrate, the animal’s energy efficiency relative to glucose is 62%, 109%, and 78%, respectively [118,122]. Accordingly, the production of acetate and butyrate results in the production of additional methanogenic substrates (formate and H_2_), which may explain the increased amount of CH_4_ emissions in high-CH_4_ emitting animals. 

## 5. Methanogenesis and Microbial Ecosystem

Several reports on the methanogenic potential of the rumen have garnered significant attention in the last decade due to the impact that methanogenesis has on ruminant animal performance and the environment [21,56,74,75,82]. Methanogens exist within several locations within the rumen, including the association with the rumen epithelium, integration into biofilms, protozoa, and fungi [21,123,124,125]. A summary of the methanogenesis and microbial fermentation of dietary components in the rumen resulting in the production of VFA, CH_4_, CO_2,_ and H_2_ produced through belching is presented in Figure 6. It has been noted that feeding concentrate diets that are high in energy substrates (non-structural carbohydrates) instantly lowered CH_4_ emission (g/d and g/kg DMI); whereas high fiber diets (forages) resulted in increased CH_4_ emissions. Ruminal methanogens utilize reducing equivalents produced by fermentative microflora (generally H_2_-producing microorganisms) such as *Ruminococcus albus*, *R. flavefaciens*, *Neocalimastrix* spp., *Desulfovibrio*, and ciliate protozoa [126,127,128,129]. According to Min et al. [4], *R. albus* and *R. flavefaciens* (cellulolytic bacteria) produced the most H_2_ among purified strains and sustained production of CH_4_ when cocultured with the *Methanobrevibacte smithii* that utilized the H_2_ to reduce CO_2_ to CH_4_ [130], which is also consistent with reports by Miller and Wolin [131] and Wolin et al. [132]. Syntrophic cooperation between H_2_ consumers (e.g., methanogens) and H_2_ producers alters the overall fermentation balance of the primary substrate toward the improved use of energy substances (Conrad et al. 1985). Subsequently, Kim et al. [133] stated that the supplementation of acetogenic bacteria (*Proteiniphilum acetatigenes*) isolated from Korean native goats (*Capra hircus coreanae*) decreased methanogenic archaea. Hence, acetogens may function as a net H_2_ sink that consequently reduces CH_4_ emissions [115].

Among the abundant bacterial phyla previously reported in numerous studies, Firmicutes and Bacteroidetes are the most abundant rumen microbiota in the guts of humans, mice, pigs, cattle, and meat goats [134,135,136,137,138,139]. Enteric CH_4_ emissions from ruminants are mainly generated by hydrogenotrophic methanogenic archaea (i.e., methanogens) that support the normal function of the rumen ecosystem through the reduction (sink) of CO_2_ by H_2_ [140,141]. Fibrinolytic bacteria, especially cellulolytic *Ruminococcus* and several *Eubacterium* spp., are well documented H_2_ producers. Conversely, the prominent cellulolytic flora, *Fibrobacter* spp., does not produce H_2_, while Bacteroidetes are net H_2_ utilizers [142]. Furthermore, the primary ciliate protozoa and fibrinolytic bacterial species in the rumen are H_2_ producing microbes that counteract CH_4_ reduction strategies that reduce available H_2_ and may slow fiber digestion [130,143]. However, the constant removal of H_2_ is vital to maintaining the biological fermentative function of the rumen because excessive H_2_ accumulation constrains carbohydrate fermentation by preventing the regeneration of NAD^+^ [140,144]. At an equivalent level of DMI, cattle diets with a higher amount of concentrate are more rapidly fermented, which results in a higher ruminal digesta passage rate, a shorter digestion time between feed particles and methanogens, and subsequently, reduced CH_4_ production and numbers of archaeal methanogens [145,146,147]. Moreover, feeding efficiently fermentable carbohydrates lowers ruminal pH and the number of cellulolytic bacteria and protozoa, resulting in reduced fiber degradation, proportionally less acetate and more propionate (thus also less free hydrogen), and, finally, less CH_4_ production, because propionate serves as an H_2_ sink [86]. A potential explanation for this could be competition for the same substrate, as *Methanobrevibacter* species are hydrogenotrophic [148] and use H_2_ and formate as substrates for CH_4_ production (Figure 6). These findings imply that the prevailing microbes in the rumen (Firmicutes and Bacteroidetes; F/B), ciliates protozoa, and methanogen archaea populations might have a role in adapting host biological parameters to reduce CH_4_ production, and can potentially be utilized to estimate CH_4_ emissions [149,150]. It has been reported that the richness of Firmicutes and the F/B ratio was positively associated with ADG due to lower A/P ratios [138,139] and positively correlated with enhanced CH_4_ emissions (Figure 5e,f [149]). These same authors confirmed that Firmicutes populations were linked to lower VFA levels when CH_4_ production was high, demonstrating that the F/B ratio could be used as an indicator to analyze rumen microbiome and GHG emissions. In addition, a significant positive relationship between fecal methanogen archaea concentration (µg/g fecal DM) and CH_4_ emissions, expressed on a DMI basis (g/kg DMI), was found (R^2^ = 0.53; n = 20) [86]. A reduction of methanogenesis or methanogens in the rumen should be associated with a decrease in methanogen archaea. 

As the single producers of CH_4_, a reasonable assumption would consider an increased abundance of methanogens within the rumen environment, producing a greater CH_4_ emission. However, the composition, rather than the abundance, of the rumen methanogen is more closely related to CH_4_ production [144]. An earlier study with 21 dairy cows fed mixed diets containing concentrate and silage showed no differences in the abundance of methanogens between high and low CH_4_-emitter dairy cows [116]. However, the same authors reported an increased relative abundance of *Methanobrevibacter gottschalkii* (1.5-fold more abundant) and *Methanobrevibacter ruminantium* (1.3-fold more abundant) that was linked with high and low CH_4_-emitting dairy cows, respectively. In addition, Lettat et al. [151] reported that CH_4_ reduction was related to the decrease in protozoa populations in multiparous dairy cattle fed different types of silage diets (corn silage vs. alfalfa silages). Correspondingly, particular species of the methanogen archaea community, rather than the overall abundance of Archaea, were found to be related to enteric CH_4_ emissions in New Zealand sheep [70,114]. However, the precise mechanism causing the high and low CH_4_ emissions phenotypes detected in sheep and cattle remains unclear [19,82,152]. Concerning the microbial community structure, previous studies reported a decrease in CH_4_ production when the archaeal richness and diversity were reduced [82,153,154]. In addition to the alterations observed within the microbiome community structure, an adaptation in the methanogenic archaeal community structure toward less efficient CH_4_-producing species is still poorly defined, and deserves further investigation.

Ciliate protozoa are important H_2_ producers that play an essential role in the interspecies H_2_ transfer and CH_4_ emissions within the rumen microbial ecosystem [155,156]. A relatively strong interaction between protozoal numbers and CH_4_ emissions has been reported and suggests that protozoa might be a good target for CH_4_ mitigation [82,156,157]. Rumen methanogen archaea can represent as much as 1–2% of the host ciliate volume [158]. Up to 20% of rumen methanogens can be found attached to protozoa [159]. In addition, dietary strategies to reduce CH_4_ by eliminating or inhibiting ciliate protozoa were reviewed by Hegarty [160] and Boadi et al. [107]. These nutritional strategies to mitigate the protozoa population included an increase in the proportion of the grain-based diet, the use of selected fatty acids (lauric- [C12:0], myristic- [C14:0] or linolenic acid [C18:3]), trace minerals (Cu and Zn), and various feed additives, such as saponins, ionophore, and monensin. Rumen ciliate protozoa are prodigious H_2_ producers, the main substrate for methanogenesis in the rumen, and their removal (defaunation; protozoa-free) yielded an average 13–45% lower CH_4_ emissions in vivo [107,155,160,161], but the results are not always consistent [141,150,162,163]. Most studies have used sheep, goat, or beef cattle as experimental models, and the effects of defaunation on the productivity of highly productive dairy cows fed intensive diets are not well known [164]. As stated in previous data [165,166,167,168], the proportion of methanogens relative to total bacteria was more evenly distributed between the liquid and solid rumen content phases in wether sheep with unaltered protozoa populations, while defaunated sheep had a lower proportion of methanogens associated with the liquid phase. These results indicate that methanogenesis is regulated not only by methanogen activity, but also impacted by various factors such as diets and varying biological ecosystems with protozoa, bacteria (Firmicutes/Bacteroidetes), and fungi community diversity affected by VFA (acetate, butyrate, and propionate), H_2_, and other substrate availability [120,149,164,165]. Therefore, future work relating to microbial diversity and the function of this community associated with animal products, especially methanogens, could be helpful to improve our understanding of the mechanisms involved in methanogenesis pathways in the rumen. In addition, cost-effective ways to change the microbial ecology to reduce H_2_ production, to re-partition H_2_ into products other than CH_4_, or to promote methanotrophic microbes with the ability to oxidize CH_4_ still need to be found and developed.

## 6. Conclusions

New technologies offer the potential to manipulate the rumen microbiome through genetic selection and varying degrees by various dietary intervention strategies to reduce CH_4_ emissions. Strategies to reduce GHG emissions, however, still need to be developed, which increase ruminant production efficiency, whereas reducing the production of CH_4_ from cattle, sheep, and goats. Many of the approaches discussed are only partial strategies; all approaches to reducing enteric CH4 emissions should consider the economic impacts on farm profitability and the relationships between enteric CH_4_ and other GHG. Numerous dietary mitigation interventions have been identified, which could help reduce CH_4_ emissions, and other strategies currently being explored and identified. The greatest declines in CH_4_ emissions are likely to be achieved through a combination of approaches, including dietary modification and improved rumen fermentation for improving feed conversion efficiency.

Dietary manipulation influences CH_4_ production by directly influencing the rumen microbiome. There is the potential to affect the rumen fermentation profiles and microbiota community structure positively and meet sustainability goals by reducing CH_4_ emissions from cattle production systems. Increased animal productivity resulted in reduced enteric CH_4_ production per animal production (milk and ADG) and improved feed efficiency. Animal DMI, GEI, ECM, ADG, and A/P ratio are the most important predictors of CH_4_ production; however, diet quality and type, rumen fermentation profiles (acetate, propionate), and microbial community structure (methanogens, bacteria, protozoa) can significantly affect this relationship. Approaches to mitigating enteric CH_4_ emissions from beef and dairy cattle production can improve animal performance and feed efficiency, while helping to reduce atmospheric GHG emissions that contribute to global warming. One possible strategy to reduce GHG emissions is a beneficial modification of the rumen microbiome to maintain a low A/P ratio and limit H_2_ production via feed management. The populations of prevailing microbial types in the rumen (Firmicutes: Bacteroidetes ratio), ciliate protozoa, and methanogen archaea might have a role in adapting host biological parameters to reduce CH_4_ production, and can potentially be utilized to estimate CH_4_ emissions. Properly designed dietary interventions can reduce enteric CH_4_ production without detrimental impacts on animal production. Therefore, GHG reduction strategies should be established to increase ruminant production efficiency, while minimizing losses of CH_4_ energy from cattle production systems. 

## Figures and Tables

**Figure 1 animals-12-00948-f001:**
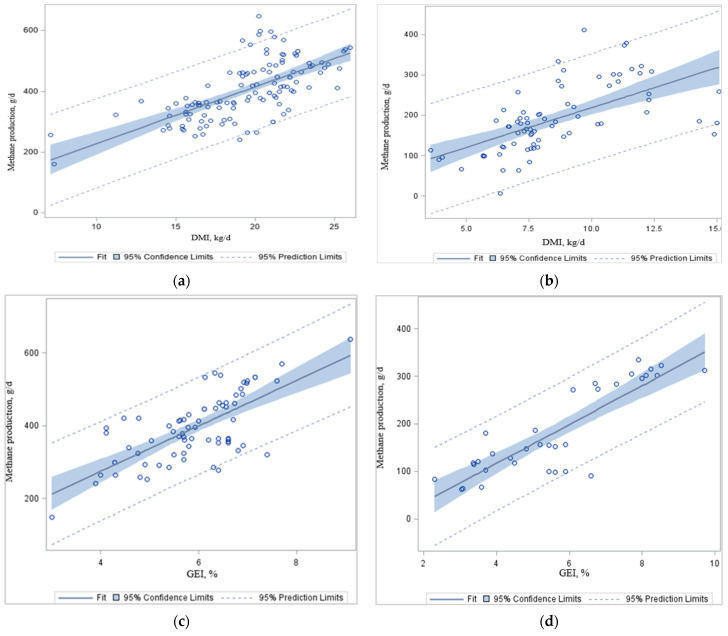
Effects of dry matter intake (DMI) and gross energy intake (GEI) on average daily methane emission (g CH_4_/d) in dairy (**a**,**c**) and beef cattle (**b**,**d**). Source: Adapted from Table 2, Table 3, Table 4, Table 5, Table 6 and Table 8. It shows the regression plots with 95% prediction and confidence limits for mean and individual predicted values of the dependent variable methane production (CH_4ⅈ_). Selected studies of methane (CH_4_) emissions associated with dry matter intake (DMI, kg/d) and gross energy intake (GEI, %).

**Figure 2 animals-12-00948-f002:**
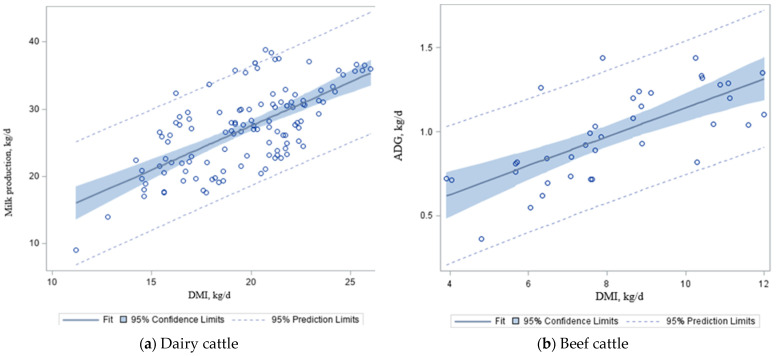
The effects of dry matter intake (DMI) on milk production (**a**) and average daily gain (ADG); (**b**) in dairy and beef cattle. Source: Adapted from Table 2, Table 3, Table 4, Table 5, Table 6 and Table 8. It shows the regression plots with 95% prediction and confidence limits for mean and individual predicted values of the dependent variables of milk production and ADG_ⅈ_.

**Figure 3 animals-12-00948-f003:**
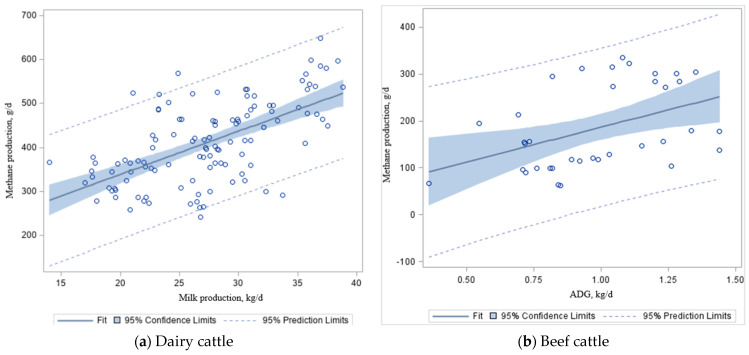
The effect of milk production (**a**) and average daily gain (ADG); (**b**) on methane (CH_4_) emissions in dairy and beef cattle fed grain-based (**c**,**e**); feedlot or dairy TMR diets) and forage-based (**d**,**f**); grazing or silage supplementation) diets, respectively. Source: Adapted from Table 2 and Table 4. It shows the regression plots with 95% prediction and confidence limits for mean and individual predicted values of the dependent variable CH_4ⅈ_.

**Figure 4 animals-12-00948-f004:**
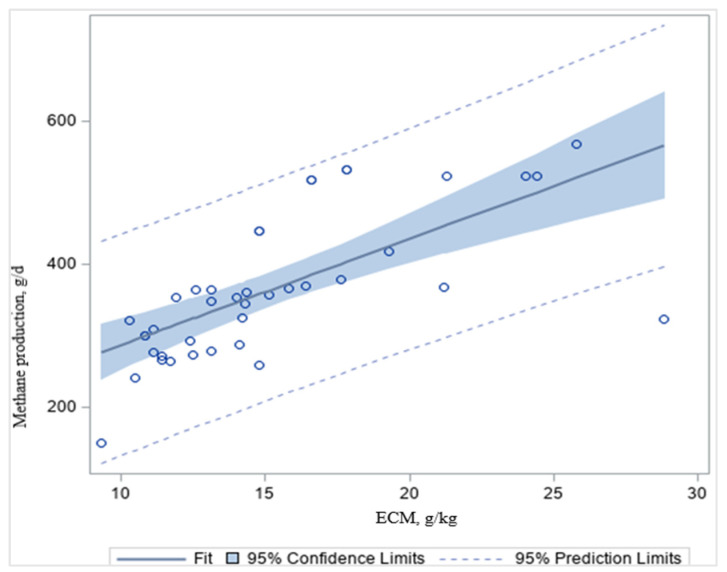
The effect of methane (CH_4_ g/d) emissions per unit of energy-corrected milk (g/kg ECM) in dairy cattle. It shows the regression plots with 95% prediction and confidence limits for mean and individual predicted values of the dependent variable. Source: Adapted from Table 6.

**Figure 5 animals-12-00948-f005:**
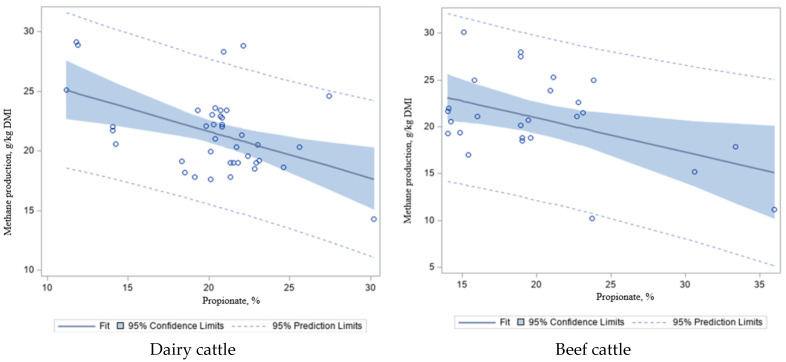
Relationship between methane (CH_4_) production and volatile fatty acids (VFA) and acetate/propionate (A/P) ratio in dairy and beef cattle. It shows the regression plots with 95% prediction and confidence limits for mean and individual predicted values of the dependent variable. Source: [14,26,27,28,29,31,33,34,35,41,44,45,46,49,50,51,52,53,54,55,56,57,58,61,62,63,64,65,66,67,69,70,71].

**Figure 6 animals-12-00948-f006:**
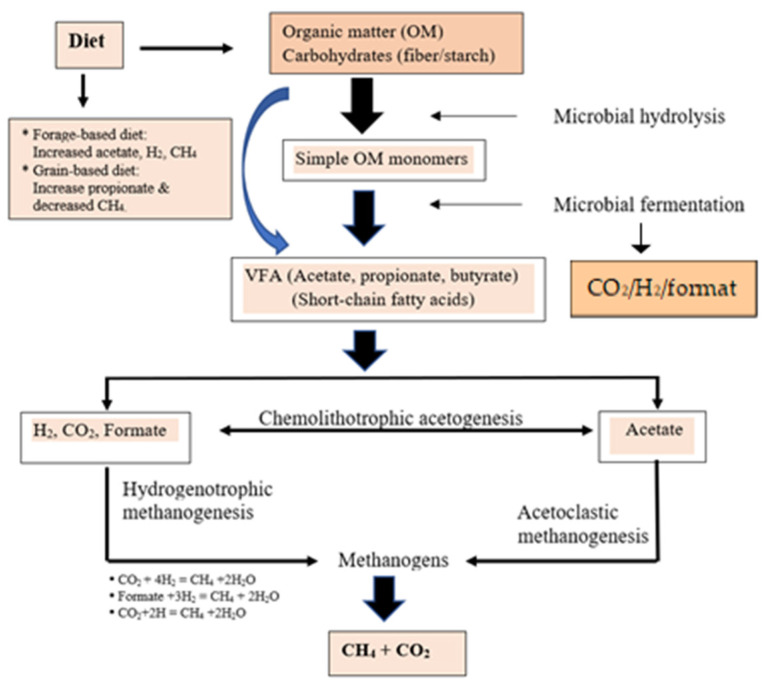
Organic matter (OM) degradation and methanogenesis pathways in the rumen under anaerobic conditions. Source: [14,17,27,32,34,42,44,45,48,52,53,55,56,59,60,65,67,71,72]. VFA = volatile fatty acids.

**Table 1 animals-12-00948-t001:** Typical composition of rumen gases.

Item	Average Percentage (%)
Hydrogen (H_2_)	0.2
Oxygen (O_2_)	0.5
Nitrogen (N_2_)	7.0
Methane (CH_4_)	20–30
Carbon dioxide (CO_2_)	45–75
Nitrous oxide (N_2_O)	minor
Hydrogen sulfate (H_2_S)	minor

Source: [4,5].

**Table 2 animals-12-00948-t002:** Enteric methane (CH_4_) emissions and milk yield (MY) from dairy cattle.

Breed	Method	Diet	No. of Animals	BW	DMI	Milk Yield (MY)	CH_4_					Ref
					kg/d	kg/d	g/d	g/kg DMI	g/kg MY	g/kg ECM	% GEI	
Holstein–Friesian	SF_6_	PRG	15	-	15	19.0	360.5	24.5	26.5	-	-	[22]
		PRG + WC	15		16.5	19.8	353.6	21.5	26	-	-	
Holstein	SF_6_	2 kg corn + grazing ^1^	10	577	14.5	19.6	287	20	15.4	14.1	-	[23]
		4 kg corn + grazing	10	552	14.2	22.4	273	19.3	12.9	12.5	-	
		6 kg corn + grazing	10	565	15.5	25.9	272	17.7	11.2	11.4	-	
		8 kg corn + grazing	10	570	15.4	26.5	277	18.1	10.8	11.1	-	
Holstein	SF_6_	0% WC	8	-	15.6	17.6	332.6	21.7	15.3	-	6.8	[24]
		15% WC	8	-	17.6	17.9	364.6	20.9	17.4	-	6.6	
		30% WC	8	-	18.6	19.3	344.2	18.6	18.5	-	5.8	
		60% WC	8	-	20.5	20.4	371.6	18.1	20.5	-	5.6	
Holstein	SF_6_	1000 kg DM/ha ^2^	23	495	16.9	22.2	286	17	13	-	5.4	[25]
		2200 kg DM/ha	23	507	15.4	21.5	286	18.7	13.6	-	6.3	
		1000 kg DM/ha	23	500	14.6	18	278	19.2	16.4	-	6.4	
		2200 kg DM/ha	23	494	14.6	17	320	22.3	19.9	-	7.4	
Holstein	RC	0% COC-oil ^3^	8	-	22.9	37.1	464	21.1	12.5	-	6.42	[26]
		1.3% COC-oil	8	-	21.4	37.5	449	21.3	11.9	-	6.35	
		2.7% COC-oil	8	-	17.9	33.7	291	17.4	8.6	-	5.19	
		3.3% COC-oil	8	-	16.2	32.4	253	16.7	7.8	-	4.94	
Holstein	SF_6_	Corn ^4^	8	537	22.2	32.1	446	20.3	-	14.8	6.12	[27]
		Wheat	8	537	21.1	32.3	300	14.3	-	10.8	4.28	
		Single-rolled barley	8	537	22.6	31.3	518	22.9	-	16.6	6.98	
		Double-rolled barley	8	537	22.7	30.6	533	23.4	-	17.8	7.15	
Holstein	SF_6_	CON	10	-	25.7	31.9	520	20.2	15.8	-	-	[28]
		Monensin ^5^	10	-	25.7	32.8	534	20.8	15.4	-	-	
		Control	10	-	23.3		433	20.2	15.2	-	-	
		Monensin	10	-	22.7		438	20.8	15.3	-	-	
		Control	10	-	20.0		429	20	13.2	-	-	
		Monensin	10	-	20.2		435	20.2	13	-	-	
		Control	10	-	20.9	32.5	466	22.5	16.5	-	-	
		Monensin	10	-	20.0	33.3	470	23.7	16.2	-	-	
Holstein	SF_6_	Low-corn ^6^	10	582	17.7	17.55	346	19.6	21	-	-	[29]
		High-corn	10	582	21.5	22.72	399	17.8	17.7	-	-	
Holstein	SF_6_	Corn ^7^	8	635	20.7	21.1	524	25.5	-	24	7.6	[30]
		Wheat	8	635	21.3	23.8	637	29.9	-	24.4	9.1	
		Corn + oil	8	635	21.7	26.1	523	24.1	-	21.3	7	
		Wheat + oil	8	635	21.8	24.9	569	26.2	-	25.8	7.7	
Holstein	RC	0% DGGS ^8^	4	700	24.2	32.6	495	20.6	15.6	-	6.09	[31]
		10% DGGS	4	701	24.6	35.1	490	20.1	14.2	-	5.8	
		20% DGGS	4	697	24.4	35.8	477	19.7	13.6	-	5.61	
		30% DGGS	4	698	25.3	36.6	475	18.9	13.2	-	5.23	
Holstein	RC	Barley control ^9^	16	616	18.7	26.6	293	16.3	17.4	12.4	4.9	[32]
		Sunflower seeds	16	623	19.5	26.7	264	14.6	17.9	11.7	4.3	
		Flaxseed	16	619	19	26.8	241	13.4	12.2	10.5	3.9	
		Canola seed	16	619	20.1	27	265	13.7	8.1	11.4	4	
Holstein	SF_6_	Corn silage-based ^10^	8	672	19.8	23	418.1	21.1	-	19.3	6.7	[33]
		Corn + CLS	8	672	19.5	21.5	369.4	18.9	-	16.4	5.7	
		Corn + ELS	8	672	16.7	20.8	258.1	15.5	-	14.8	4.8	
		Corn + LSO	8	672	14.7	18.9	149.2	10.2	-	9.3	3	
Holstein	RC	CON ^11^	10	-	16.4	28.9	362	22.1	12.8	-	6.2	[34]
		Feed additives	10	-	15.9	26.1	325	20.5	12.7	-	5.7	
		Control	6	-	20	32	-	-		-	-	
		Feed additives	6	-	19.8	33.2	-	-		-	-	
Holstein	RC	47 Forage: 53 Conc ^12^	8	546	20.7	38.8	538	25.9	14	-	-	[14]
		54 forage: 46 Conc	8	546	21.0	38.4	597	28.2	15.9	-	-	
		61 forage: 39 Conc	8	546	20.2	36.9	586	29.1	16.1	-	-	
		68 Forage: 32 Conc	8	546	20.2	36.9	648	31.9	17.8	-	-	
Jersey	SF_6_	Grasses	9	480	15.6	20.5	325	20.7	14.9	14.2	-	[35]
		Legumes	9	480	16.5	22	278	17.4	14.7	13.1	-	
		Forbes	9	480	17	22.9	348	20.2	14.7	13.1	-	
Holstein	RC	Low^13^-intake 1	7	-	15.8	25.1	308	19.7	12.3	11.1	5.7	[36]
		Low-intake 2	7	-	15.7	22.6	353	22.6	16.1	14	6.6	
		Low-intake 3	7	-	16	22.1	357	22.2	16.3	15.1	6.6	
		Low-intake 4	7	-	14.5	20.9	345	24.3	16.8	14.3	6.9	
		High-intake 1	7	-	16.8	29.5	321	19.3	11.1	10.3	5.5	
		High-intake 2	7	-	16.4	27.6	354	21.4	12.9	11.9	6.4	
		High-intake 3	7	-	16.9	28.5	365	21.7	12.8	12.6	6.4	
		High-intake 4	7	-	16.2	28	364	22.8	13.2	13.1	6.6	
Holstein	RC	Grass silage	6	132.5	17.8	22.01	365.5	20.6	17.6	15.81	5.86	[37]
		Sainfoin silage	6	132.5	18.7	24.08	360.8	19.4	15.5	14.36	5.71	
Jersey	SF_6_	CON	11	385	11.2	9.03	323	29.1	35.5	28.8	-	[38]
		4 kg Conc	11	389	12.8	14	367	28.9	25.1	21.2	-	
		8 kg Conc	11	388	15.6	17.7	378	25.1	21.1	17.6	-	
Holstein	GF	High-CS ^14^	10	677	25.2	35.6	410	16.	11.7	-	-	[39]
	RC	High-CS + NDF	10	677	24.1	33.3	461	18.9	14.2	-	-	
		High-GS	10	665	19.5	30	460	24	15.6	-	-	
		High-GS + NDF	10	661	19	28	460	24.1	16.4	-	-	
		High-CS	4	693	21.7	32.9	495	21.8	15.6	-	-	
		High-CS + NDF	4	688	20.5	30.7	472	23.7	15.8	-	-	
		High-GS	4	664	18.4	29.5	462	25.5	15.4	-	-	
		High-GS + NDF	4	676	17	27.1	418	24.2	16.3	-	-	
Holstein	RC	CON	6	626.5	21.8	30.5	416.8	19.2	-	-	5.7	[40]
		Yucca	6	629.6	22	31	415.4	19	-	-	5.63	
		Quillaja	6	625.8	21.2	30.3	384.9	18.5	-	-	5.48	
	SF_6_	Control	6	626.5	21.8	30.5	325.3	16.1	-	-	4.76	
		Yucca ^4^	6	629.6	21.5	31	359	17	-	-	5.03	
		Quillaja ^4^	6	625.8	22.1	30.3	339	15.4	-	-	4.57	
Holstein	RC	Corn silage (CS) ^15^	4	643.4	20.3	36.1	598	29.5	16.5	-	-	[41]
		CS + linseed oil	4	643.4	21.2	37.4	580	27.4	15.5	-	-	
		Grass silage (GS)	4	643.4	19.2	35.7	567	29.5	16.1	-	-	
		GS + linseed oil	4	643.4	19.7	35.4	553	28.1	15.7	-	-	
Holstein	RC	Grazing	7	341	18.4	19.06	309	16.7	16.2	-	-	[42]
		Monensin	7	365	18.0	19.51	306	17	15.7	-	-	
Holstein	SF_6_	Control	12	614.6	22.6	27.2	400	17.8	14.8	-	5.4	[43]
		Almond hull	10	614.6	22.6	24.5	430	19.1	17.7	-	5.8	
		Citrus pulp	10	614.6	21.0	26.1	414	19	16.6	-	6	
Holstein	RC	CS ^16^, 49.3%	8	608	20.3	27	378	18.6	14.4	-	5.67	[44]
		AS, 26.8%	8	608	20.9	27.3	396	19	14.8	-	5.92	
		WS, 20%	8	608	20.9	28.2	396	19	14.4	-	5.78	
		Hay-based, 25.3%	8	608	23.4	29.3	413	17.8	14.2	-	5.59	
Holstein	RC	Control	9	660	21.3	14.8	539	21.3	14.8	-	6.44	[45]
		Ground Feba bean	9	660	20.3	15	533	20.3	15	-	6.13	
		Rolled Feba bean	9	660	21	15.2	544	21	15.2	-	6.33	
Holstein	RC	CON ^17^	4	541	19.2	27.8	461	22.8	-	-	6.73	[46]
		Low- oregano	4	541	19.4	29.8	455	22	-	-	6.49	
		Medium- oregano	4	541	19.9	29.9	464	22.2	-	-	6.56	
		High- oregano	4	541	19.2	28	451	22.2	-	-	6.56	
Holstein	RC	CON	4	712	21.7	24.1	502	23.4	-	-	6.87	[46]
		Low- oregano	4	712	20.9	23.2	487	23.4	-	-	6.89	
		Medium- oregano	4	712	21.8	23.3	520	23.6	-	-	6.92	
		High- oregano	4	712	21.3	23.2	485	23	-	-	6.76	
Holstein	GF	CON	10	-	22.5	28.2	525	23.5	-	-	-	[47]
		3-NOP + hay	10	-	21.3	26.7	380	18.1	-	-	-	
		3-NOP + Conc	10	-	22.3	28	403	18.6	-	-	-	
		Control	10	-	23.4	31.3	494	21.5	-	-	-	
		3-NOP + hay	10	-	23.6	31	486	20.7	-	-	-	
		3-NOP + Conc	10	-	23.5	32.8	482	20.8	-	-	-	
		Control	10	-	20.9	25	464	21.8	-	-	-	
		3-NOP + hay	10	-	21.2	22.7	427	20.2	-	-	-	
		3-NOP + Conc	10	-	22.4	25.2	464	21.2	-	-	-	
Jersey	GF	CON ^18^	4	-	18.2	19.8	362.6	19.9	-	-	-	[48]
		CON + yeast	4	-	18.6	20.8	364.2	19.6	-	-	-	
		NO3	4	-	17.2	19.6	303.2	17.6	-	-	-	
		NO3+ yeast	4	-	16.6	19.3	301.6	18.2	-	-	-	
Holstein/	RC	CON ^19^	4	508.1	19.1	26.3	421.6	22.3	-	-	-	[49]
Jersey		DGGS	4	513.4	20.1	27.5	421.9	21.4	-	-	-	
		DGGS+ corn oil	4	513.2	20	28.3	384.7	19.9	-	-	-	
		DGGS+ CaS	4	510.7	19.6	27.6	381.4	19.6	-	-	-	
No. of Observation			127									

BW = body weight; COC = coconut; CON = control; Conc = concentrate; DGGS = dried distillers’ grains solubles; DMI = dry matter intake; ECM= energy-corrected milk; GEI = gross energy intake; GF= GreenFeed system (C-Lock, ND); MF = milk fat; MP = milk protein; MS = milk solid; MY = milk yield; N = number of animals; RC: open-circuit respiration chamber; PRG = perennial rye grass; Ref = reference; SF_6_ = sulfur hexafluoride; WC = white clover; 3-nitrooxypropanol (3-NOP). ^1^ The effect of concentrate (Conc) feed level (2.0, 4.0, 6.0, and 8.0 kg/cow per day; fresh basis) on enteric CH_4_ emissions from cows grazing perennial ryegrass-based swards; ^2^ 1000 kg of dry matter (DM)/ha (low herbage mass, LHM) or 2200 kg of DM/ha (high herbage mass, HHM); ^3^ Diets differed in concentrations of coconut (COC) oil: 0.0 (control) or 1.3, 2.7, or 3.3% COC, DM basis; ^4^ Offered 1 of 4 diets: corn diet of 10.0 kg of DM/d of single-rolled corn grain, 1.8 kg of DM/d of canola meal, 0.2 kg of DM/d of minerals, and 11.0 kg of DM/d of chopped alfalfa hay; a wheat diet (WHT) similar to the corn diet but with the corn replaced by single-rolled wheat; a barley diet (SRB) similar to the corn diet but with the corn replaced by single-rolled barley; and a barley diet (DRB) similar to the corn diet but with the corn replaced by double-rolled barley; ^5^ Monencin = 471 mg/cow/d on top-dressed on 4 kg (DM)/d of rolled barley grain offered in a feed trough twice daily at milking times; ^6^ The two levels of concentrate supplementation (1 vs. 6 kg/animal daily) were randomly allocated within blocks, giving 12 animals per treatment; ^7^ The corn diet included 8.0 kg of DM/d of crushed corn grain, the wheat diet (WHT) included 8.0 kg of DM/d of crushed wheat grain, the corn plus fat diet (CPF) included 8.0 kg of DM/d of crushed corn grain and 0.80 kg/d of canola oil, and the wheat plus fat diet (WPF) included 8.0 kg of DM/d of crushed wheat grain and 0.80 kg/d of canola oil; ^8^ The dietary treatments were: (1) 0% dried distillers’ grains solubles (DDGS), (2) 10% DDGS, (3) 20% DDGS, and (4) 30% DDGS, on a DM basi; ^9^ The dietary treatments were: (1) a commercial source of calcium salts of long-chain fatty acids (CTL), (2) crushed sunflower seeds (SS), (3) crushed flaxseed (FS), and (4) crushed canola seed (CS). The oilseeds added 3.1 to 4.2% fat to the diet (DM basis); ^10^ A control diet (CON) based on corn silage (59%) and concentrate (35%), and the same diet supplemented with whole crude linseed (CLS), extruded linseed (ELS), or linseed oil (LSO) at the same fatty acids (FA) level (5% of dietary DM); ^11^ The mixture of feed additives contained lauric acid, myristic acid, linseed oil, and calcium fumarate. These additives were included at 0.4, 1.2, 1.5, and 0.7% of dietary DM, respectively; ^12^ Concentrate:forage ratio: 47:53, 54:46, 61:39, and 68:32, DM basis. Forage consisted of alfalfa silage and corn silage in a 1:1 ratio; ^13^ Diets contained grass silage, corn silage, and a compound feed meal was 70:10:20% on a DM basis, respectively. Treatments consisted of 4 grass silage qualities prepared from a grass harvested from leafy through the late heading stage and offered to dairy cows; ^14^ High corn silage (CS) versus high grass silage (GS), without or with added neutral detergent fiber (NDF); ^15^ Diets contained 500 g of forage/kg of DM containing corn silage (CS) and grass silage (GS) in proportions (DM basis) of either 75:25 or 25:75 for high CS or high GS diets, respectively. Extruded linseed supplement (275 g/kg ether extract, DM basis) was included in treatment diets at 50 g/kg of DM.; ^16^ Corn silage (CS), alfalfa silage (AS), wheat silage (WS), and a typical hay-based diet (alfalfa/Italian ryegrass hays) were used; ^17^ Experiment 1 used low essential oil (EO) oregano (0.12% EO of oregano DM) and evaluated a control (C) diet with no oregano and 3 oregano diets with 18 (low; L), 36 (medium; M), and 53 g of oregano DM/kg of dietary DM (high; H). Experiment 2 used high EO oregano (4.21% EO of oregano DM) with 0, 7, 14, and 21 g of oregano DM/kg of dietary DM for C, L, M, and H, respectively. Oregano was added to the diets by substituting grass/clover silage on a DM basis; ^18^ Diets containing either urea or 1.5% NO_3_− (DM basis; isonitrogenous to control) and without or with *Saccharomyces cerevisiae* (Alltech Inc.); ^19^ Treatments were composed of control (CON) diet, which did not contain reduced-fat distiller’s grain and solubles (DDGS), and treatment diets containing 20% (dry matter basis) DDGS (DG), 20% DDGS with 1.38% (dry matter basis) added corn oil (CO), and 20% DDGS with 0.93% (DM basis) added calcium sulfate (CaS); Source: [14,22,23,24,25,26,27,28,29,30,31,32,33,34,35,36,37,38,39,40,41,42,43,44,45,46,47,48,49].

**Table 3 animals-12-00948-t003:** The ordinary least squares regression (OLS) estimates of milk production (a) and dry matter intake (DMI) impacts on methane production (CH_4_) in dairy and beef cattle production, and dairy and beef cattle fed grain-based and forage-based diets.

	Model 1	Model 2	Model 3	Model 4	Model 5	Model 6
	Dairy Cattle	Beef Cattle	Dairy Cattle; Grain-Based	Dairy Cattle; Forage-Based	Beef Cattle; Grain-Based	Beef Cattle; Forage-Based
Variable	CH_4_ Production	CH_4_ Production	CH_4_ Production	CH_4_ Production	CH_4_ Production	CH_4_ Production
Milk Production	9.82	-	3.14	6.54	-	-
	(*p* < 0.001)		(*p* = 0.12)	(*p* < 0.01)		
ADG	-	117.33	-	-	151.26	143
		(*p* < 0.01)			(*p* < 0.01)	(*p* < 0.01)
Intercept	142.69	38.34	327.09	22.91	11.29	49.01
R-Square	37.15%	18.90%	4.17%	11.08%	38.03%	40.04%
Number of obs.	115	36	58	55	18	17
Parameters	2	2	2	2	2	2
MSE	5418.6	6705.8	9523.6	7675.2	2491.3	4216.6

Note: Obs. = observations. ADG = average daily gain; CH_4_ = methane; MSE = mean squared errors.

**Table 4 animals-12-00948-t004:** Enteric methane (CH_4_) emissions and animal performance from beef cattle.

Item	Method	Experimental Diet	No. of Animal	Initial BW	ADG kg/d	DMI kg/d	CH_4_				Ref
Breed							g/d	g/kg DMI	g/kg ADG	% GEI	
Hereford + Simmental	SF_6_	78% AL + 22% MB	16	511.2	-	11.4	378.8	33.23	-	7.1	[50]
(heifers)		100% MB	16		-	9.7	411	42.37	-	9.5	
Brahman heifers	RC	AG grass ^1^	6	353	-	3.58	113	31.5	-	1.9	[51]
		RG grass	6	364	-	7.07	257	36.3	500.4	2.07	
		Grain + AL	6	380	-	7.31	160	21.9	127.3	1.23	
Holstein steers	RC	Forage-based ^2^	8	311.6	-	7.4	166.2	22.64	-	6.47	[52]
		Proteolytic enzyme	8	311.6	-	7.55	164.4	22.11	-	6.32	
		Monensin	8	311.6	-	7.71	159.6	20.7	-	5.91	
		Sunflower oil	8	311.6	-	6.91	129	18.81	-	5.08	
Holstein steers	RC	Forage-based ^3^	8	311.6	-	7.18	267	25.05	-	7.13	[52]
		Fumaric acid	8	311.6	-	6.69	250	26	-	7.4	
		Levucell yeast	8	311.6	-	6.71	243	26.43	-	7.53	
		Procreatin yeast	8	311.6	-	7.46	272	24.32	-	6.93	
Crossbreed	SF_6_	New breed-grazing	20	275	0.692	6.49	213	32.8	0.324	-	[53]
(Charolais × Zebu)		Cross line-grazing ^4^	13	287	0.62	6.36	-	-	-	-	
		Old-breed-grazing	13	282	0.547	6.06	194	32	0.337		
Crossbreed	GF	New breed-feedlot		379	1.44	10.25	178	17.36	0.149	5.19	[53]
(Charolais × Zebu)		Cross-breed-Feedlot		383	1.32	10.42	-	-	-	-	
		Old breed-Feedlot		362	1.23	9.11	156	17.12	0.124	5.07	
Crossbreed	RC	TMR ^5^	40	357	0.187	6.2	187	30.4	0.52		[54]
Crossbreed steers	SF_6_	CON ^6^	25	292	0.716	7.01	151.5	22	0.21	-	[55]
		CT + high forage	25	293	0.733	7.27	156.4	21.7	0.21	-	
		HT + high forage	25	292	0.715	7.52	155	20.7	0.22	-	
Angus heifers and steers	SF_6_	CON ^7^	12	255	0.81	5.68	98.7	18.82	0.39	5.61	[56]
		1% CT DM	12	254	0.82	5.72	99.1	18.51	0.39	5.9	
		2% CT DM	12	255	0.76	5.67	99.7	18.79	0.39	5.45	
Nellore steers	SF_6_	CON ^8^	9	419	1.15	8.88	147	17.1	0.35	4.81	[57]
		Palm oil	9	404	0.36	4.8	66.8	9.55	0.16	3.59	
		Linseed oil	9	416	0.85	7.1	62.8	12.5	0.15	3.05	
		Protected fat	9	434	0.99	7.57	118	15.9	0.27	4.5	
		Whole soybean	9	434	0.84	6.47	63.9	12.7	0.15	3.07	
Nellore Bulls	SF_6_	High-starch + CG ^9^	9	239.45	0.89	7.7	117.74	15.36	0.492	3.37	[58]
		High-starch - no CG	9	259.11	1.03	7.69	127.63	17.14	0.493	4.38	
		Low-starch + CG	9	257.55	0.92	7.45	114.61	15.45	0.445	3.39	
		Low-starch + no CG	9	246.66	0.97	7.85	120.48	15.44	0.488	3.49	
Crossbreed steers	SF_6_	CS (09/13)	12	530	1.28	10.88	301	29.4	0.568	8.4	[59]
		CS (09/28)	12	531	1.35	11.95	304	25.8	0.582	7.7	
		Corn silage (10/09)	12	531	1.2	11.13	301	27.7	0.56	8.1	
		CS (10/23)	12	531	1.29	11.08	284	26.2	0.53	7.3	
Crossbreed steers	SF_6_	WS-1	18	539	0.82	10.3	195	30.1	0.547	8	[60]
		WS-2	18	539	1.04	11.6	315	27.5	0.584	8.24	
		WS-3	18	538	1.103	12	322	28	0.598	8.52	
		WS-4	18	538	1.043	10.7	273	25	0.507	6.79	
		GS	18	439	0.929	8.9	312	35.6	0.711	9.72	
		Conc	18	537	1.335	10.4	180	15.3	0.335	3.71	[61]
Crossbreed	SF_6_	CON	12	338	1.44	7.88	137.8	17.9	0.408	3.9	
(Charolais x Limousin)		Whole soybean	12	338	1.26	6.32	103	15.2	0.304	3.7	
		Refined soy oil	12	338	1.55	7.52	83.9	11.2	0.248	2.3	
Cross breed	SF_6_	CON	12	474	1.08	8.67	334.4	38.8	0.243	7.9	[62]
Charolais x Limousin)		Refined coconut oil	12	474	1.24	8.81	271.6	31.1	0.168	6.1	
		Copra meal	12	474	1.2	8.66	284.6	33.2	0.192	6.7	
Holstein steers/heifers	RC	Steer ^10^	10	175	0.71	4.04	96.4	23.8	2.1	-	[63]
		Heifer	10	176	0.72	3.91	90.5	23.2	1.88	-	
Crossbreed beef heifers	RC	CON ^11^	8	388.5	-	9.05	228	25.3	0.065	7.8	[64]
		CDDGS	8	388.5	-	8.57	184	21.5	0.055	6.6	
		WDDGS	8	388.5	-	8.13	191	23.9	0.061	7.3	
		WDGGS + corn oil	8		-	8.42	174	21.1	0.054	6.3	
Holstein heifers	RC	CON (Grass hay + Conc; 50:50%) ^12^	4	656.3	-	12.4	308.6	25	0.038	7.2	[65]
(non-lactating)		CON + 4% LO	4	656.3	-	12.3	238.1	19.4	0.0296	5.8	
		CON + 3% calcium nitrate	4	656.3	-	12.3	252.7	20.7	0.031	5.6	
		CON + 4% LO + 3% nitrate	4	656.3	-	12.2	206.8	17	0.026	4.8	
Beef cattle	SF_6_	Grazing 1 cow/ha	12	526.2	-	11.3	372.7	26.2	-	8.4	[66]
(Cannulated Angus)		Grazing 2.5 cow/ha	12	529.5	-	15	181.5	11.3	-	3.7	
		Grazing 1 cow/ha	12	550.7	-	15.1	258.6	16.1	-	5	
		Grazing 2.5 cow/ha	12	558.6	-	14.9	143.6	10.8	-	3.2	
		Grazing 1 cow/ha	12	563.9	-	14.3	185.7	16.8	-	3.1	
		Grazing 2.5 cow/ha	12	559.4	-	15.3	158.7	10.7	-	3.3	
		Grazing 1 cow/ha	12	578.3	-	17.9	176.1	9.6	-	5.3	
		Grazing 2.5 cow/ha	12	570.8	-	17.7	275.1	14.8	-	4.8	
Angus heifers	RC	CON	12	255	0.81	5.68	98.7	18.82	-	5.61	[17]
		1% CT	12	254	0.82	5.72	99.1	18.51	-	5.9	
		2% CT	12	255	0.76	5.67	99.7	18.9	-	5.45	
Limousin cross heifers	SF_6_	Low-forage mass	15	346	-	6.5	120	19.3	0.135	5.6	[67]
		High-forage mass	15	346	-	6.44	122	21.1	0.163	6.1	
Holstein growing heifers	RC	High-CS ^13^	4	454	-	9.29	220	22.3	-	-	[68]
		High-CS + LO	4	454	-	9.46	197	20.4	-	-	
		High-GS	4	448	-	7.94	203	27	-	-	
		High-GS + LO	4	447	-	7.89	201	26.2	-	-	
		High-CS	4	361	-	7.03	184	26.1	-	-	
		High-CS + LO	4	364	-	7.16	193	27	-	-	
		High-GS	4	361	-	7.28	208	28.5	-	-	
		High-GS + LO	4	365	-	7.42	192	26	-	-	
No. of observations			82								

AL = alfalfa (*Medicago sativa*); BW = body weight; CON = Control; Conc = concentrate; CS = corn silage; CT = condensed tannins; DGGS = Dried distillers’ grains solubles; DMI = dry matter intake; CG= crude glycerin; GEI = gross energy intake; GF = GreenFeed system (C-Lock, ND); GS= grass silage; HT = hydrolysable tannins; LO = linseed oil; MB = meadow bromegrass (*Bromus biebersteinii*); N = number of animal; RC: open-circuit respiration chamber; PRG = perennial rye grass; Ref = reference; SF_6_ = sulfur hexafluoride; TMR = total mixed ration; WC = white clover; WS= wheat silage; ^1^ Angleton grass (AG), Rhodes grass (RG), alfalfa (AL), and a high-grain diet; ^2^ Proteolitic enzyme (1 mL/kg DM), Monensin (33 mg/kg DM), and sunflower oil (400 g/d); ^3^ Treatments were control (no additive), procreatin-yeast (4 g/d), Levucell SC yeast (1 g/d), and fumaric acid (80 g/d); ^4^ Canchim steers from three different lines (5/8 Charolais x 3/8 Zebu) were used: old, new, and their cross; ^5^ TMR diet including lucerne and oaten hay chaff; ^6^ A basal diet of alfalfa, barley silages (50:50; dry matter [DM] basis) and supplemented with hydrolyzable tannins (HT) extract (chestnut) or a combination (50:50) of HT and condensed tannins (CT) extracts (quebracho CT); ^7^ Three treatments at 0, 1, and 2% of dietary DM as CT extracts; ^8^ Without fat (WF), palm oil (PO), linseed oil (LO), protected fat (PF), and whole soybeans (WS); ^9^ Starch-based supplementation level combined with crude glycerin (CG); ^10^ TMR diet with grass silage and concentrates (0.45 and 0.55, DM basis, respectively); ^11^ Control diet contained 55% whole crop barley silage, 35% barley grain, 5% canola meal, and 5% vitamin and mineral supplement. Three dried distillers’ grains solubles (DDGS) diets were formulated by replacing barley grain and canola meal (40% of the dietary DM) with corn-based DDGS (CDDGS), wheat-based WDDGS, or WDDGS plus corn oil (WDDGS + oil). For the WDDGS+ oil treatment, corn oil was added to WDDGS in a ratio of 6:94 to achieve the same fat level as in CDDGS; ^12^ Control (1) (CON; 50% natural grassland hay and 50% concentrate), (2) CON with 4% linseed oil (LIN), (3) CON with 3% calcium nitrate (NIT), and (4) CON with 4% linseed oil plus 3% calcium nitrate (LIN + NIT); ^13^ TMR diet with forage containing high corn silage (CS) or high grass silage (GS) and concentrates in proportions (forage: concentrate, DM basis) of either 75:25 (experiment 1) or 60:40 (experiment 2), respectively; Source: [17,50,51,52,53,54,55,56,57,58,59,60,61,62,63,64,65,66,67,68].

**Table 5 animals-12-00948-t005:** The ordinary least squares regression (OLS) estimates of dry matter intake (DMI) impacts on milk production and on average daily gain (ADG) in dairy and beef cattle production, respectively.

	Model 1	Model 2
	Dairy Cattle	Beef Cattle
Variable	Milk Production	ADG
DMI	1.31	0.09
	(*p* < 0.001)	(*p* < 0.01)
Intercept	1.34	2.44
R-Square	44.44%	50.17%
Number of observations	118	38
Parameters	2	2
MSE	19.958	0.0368

DMI = dry matter intake; ADG = average daily gain; MSE = mean squared errors.

**Table 6 animals-12-00948-t006:** The ordinary least squares regression (OLS) estimates of methane (CH_4_ g/d) emissions per unit of energy-corrected milk (g/kg ECM) on methane production (CH_4ⅈ_) in dairy cattle.

	Model 1
Variable	Dairy Cattle
	Methane (CH_4_) Production
ECM	9.82
	(*p* < 0.001)
Intercept	138.95
R-Square	45.98%
Number of observations	40
Parameters	2
MSE	5570.2

ECM = energy-corrected milk (g/kg ECM); MSE = mean squared errors.

**Table 7 animals-12-00948-t007:** The ordinary least squares regression (OLS) estimates of propionate, acetate, and acetate/propionate (A/P) impacts on methane (CH_4_) production in dairy and beef cattle.

	Model 1	Model 2	Model 3	Model 4	Model 5	Model 6
	Dairy Cattle	Beef Cattle	Dairy Cattle	Beef Cattle	Dairy Cattle	Beef Cattle
Variable	CH_4_ Production (DMI)	CH_4_ Production (DMI)	CH_4_ Production (DMI)	CH_4_ Production (DMI)	CH_4_ Production	CH_4_ Production
Propionate %	−0.55 ***	−0.4 **				
	(*p* < 0.001)	(*p* < 0.02)				
Acetate %			0.87 ***	0.48 ***		
			(*p* < 0.001)	(*p* < 0.01)		
A/P ratio					0.28 ***	0.09 **
					(*p* < 0.001)	(*p* < 0.01)
Intercept	32.06	32.43	4.08	7.31	15.5	15.01
R-Square	21.41%	21.35%	27.63%	10.35%	45.07%	14.52%
No. of Obs	40	26	39	26	37	26
Parameters	2	2	2	2	2	2
MSE	8.8428	17.399	7.2949	19.833	4.8736	18.911

Note: A/P ratio = acetate/propionate ratio; DMI = dry matter intake; Methane = CH_4_; *p*-values in parentheses *** *p* < 0.001, ** *p* < 0.01. No. of Obs. = number of observations; MSE = mean squared errors.

**Table 8 animals-12-00948-t008:** The ordinary least squares regression (OLS) estimates of animal performance impact on methane production (CH_4_) in dairy and beef cattle production.

	Model 1	Model 2	Model 3	Model 4
	Dairy Cattle	Beef Cattle	Dairy Cattle	Beef Cattle
Variable	CH_4_ Production	CH_4_ Production	CH_4_ Production	CH_4_ Production
DMI	18.53	18.93	-	-
	(*p* < 0.001)	(*p* < 0.001)		
GEI	-	-	62.2	40.93
			(*p* < 0.001)	(*p* < 0.001)
Intercept	42.37	22.33	27.76	47.16
R-Square	44.42%	36.61%	49.92%	74.10%
No. of Obs	121	74	72	34
Parameters	2	2	2	2
MSE	5113.5	4425.8	4418.1	2286.8

Note: Obs. = observations; DMI = dry matter intake; DEI = gross energy intake; MSE = mean squared errors.

## Data Availability

Not available.

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
