# Peer review of "Enteric Methane Emissions and Animal Performance in Dairy and Beef Cattle Production: Strategies, Opportunities, and Impact of Reducing Emissions"

_animals, 2022, doi:10.3390/ani12080948_

Round 1

Reviewer 1 Report

I think it is well organized, but there is nothing new in the content considering that it is a review article.

The coefficient of determination (R2) is often a low value. Isn't the fact that it is significant (P<0.05) an indication that the correlation is significantly low (R=near zero, P<0.05)? The regression equation doesn't fit well ?
L174; R2 = 0.34
L207; R2 = 0.37
L208; R2 = 0.19
L212; R2 = 0.38-0.40
L277; R2= 0.21
L279; R2 = 0.28, R2 = 0.10
L280; R2 = 0.45-0.15
Table 6. R-Square; Model 2, 18.90%; Model 3, 4.17%; Model 4, 11.08%
Table8. R-Square; 10.35% - 45.07%

My specific comments are as follows,

L77; energy corrected milk (ECM) --- You should show how you calculated it.
L131; Recently, Min et al. [32] indicated ... L134; Min et al. [32], the average estimate ... --- In the end, are these three methods more or less accurate, or do they show the same value?
L147; dry matter (DM) --- DM
L181; kilogram --- kg ?
L239; energy corrected milk (ECM) --- ECM
L288; and other thermophilic bacterial species [65, 66, 67, 68] --- Not in italics.
L367; positively correlated with enhanced CH4 emissions --- Is this logically correct within this sentence?
L456; J. Dairy Sci. --- italic, All subsequent Journals' names should be changed.
L843; respiration chamber (RC); --- respiration chamber;
L892-893; dry matter basis --- DM basis
L902; respiration chamber (RC); --- respiration chamber;

Figure 6. CO2+4H2=CH4+2H2O, CO2+2H=CH4+2H2O, This begs the question, where did the end product CO2 come from?

Author Response

Reviewer 1

The coefficient of determination (R2) is often a low value. Isn't the fact that it is significant (P<0.05) an indication that the correlation is significantly low (R=near zero, P<0.05)? The regression equation doesn't fit well ?
L174; R2 = 0.34
L207; R2 = 0.37
L208; R2 = 0.19
L212; R2 = 0.38-0.40
L277; R2= 0.21
L279; R2 = 0.28, R2 = 0.10
L280; R2 = 0.45-0.15
Table 6. R-Square; Model 2, 18.90%; Model 3, 4.17%; Model 4, 11.08%
Table8. R-Square; 10.35% - 45.07%

Answer: "P<0.05" indicates that the estimated coefficients are significantly different from zero at a 5% significance level, the independent variable has a significant effect on the dependent variable. The smaller the P-value, the stronger evidence that the coefficient of the estimates significantly differs from zero.

In this project, R-square values are between 10.35%-45.05%, which indicates the percentage of the variance in the dependent variable that is explained by the independent variables. The regression test's power is measured by F statistics (or the MSE). A significant F-test indicates that the observed R-squared is reliable. For the regression with a low R-squared value, but the independent variables are statistically significant, we still can make conclusions about the relationship between the variable of the interest and the dependent variable. Even though R-square could be improved if other variables affecting the dependent variables were included, it is not necessary or feasible for this project since our data is from the previous studies and other factors are limited or unavailable.

My specific comments are as follows,

L77; energy corrected milk (ECM) --- You should show how you calculated it. We didn’t calculate this value. We just used the ECM value from the published data.
L131; Recently, Min et al. [32] indicated ... L134; Min et al. [32], the average estimate ... --- In the end, are these three methods more or less accurate, or do they show the same value? Deleted the 2nd Min et al., (32).
L147; dry matter (DM) --- DM: changed to DM.
L181; kilogram --- kg ? Changed to kg,

L239; energy corrected milk (ECM) --- ECM: changed to ECM.
L288; and other thermophilic bacterial species [65, 66, 67, 68] --- Not in italics. Changed to no italic:  other thermophilic bacterial species [65, 66, 67, 68]
L367; positively correlated with enhanced CH4 emissions --- Is this logically correct within this sentence? Changed to “were positively associated with ADG”
L456; J. Dairy Sci. --- italic, All subsequent Journals' names should be changed. Changed to Italic”

L843; respiration chamber (RC); --- respiration chamber; Changed to respiration chamber.
L892-893; dry matter basis --- DM basis: changed to DM basis.
L902; respiration chamber (RC); --- respiration chamber; Changed to respiration chamber.

Figure 6. CO2+4H2=CH4+2H2O, CO2+2H=CH4+2H2O, This begs the question, where did the end product CO2 come from?

Added origin of CO2/H2/formate in the figure.

Reviewer 2 Report

General comments: 

The manuscript "Enteric methane emissions and animal performance in dairy and beef cattle production: enumerating  the opportunities and impact of reducing emissions" is an interesting paper about methane production from livestock production and provide an overview about the biochemical process in the rumen and possible impact for a reduction in methane production. 

The title is very similiar to the paper from Knaap et al. 2014: Enteric methane in dairy cattle production: quantifying opportunities and impact of reducing emissions. Was that desired from you?

Overall comment: In your title you would like to enumerate the opportunities and impact of reducing emissions but that is not very clear described in the manuscript. Please add one part where AU describe relevant methods and out of an practical point of view of reducing methane emissions from dairy and beef cattle.

Ln 85: You mentioned that in a database a lot of studies examined the effects of mitigation strategies on enteric CH4 emissions..... Could you please cite the studies which were used.

110: Is it true that 1033 studies are available?

Ln 131-133: You only describe the CH4-measurement methods RC, SF6 and GF but you do not describe the methane measurement with a laser detector. Why do you not describe or mention that measurement method because there are some paper about that measurement method in the literature available.

Ln 139: What do you mean by animal intake? Do you mean feed intake? Please change.

Ln 148: CH4 has an energy content of 55.65 MJ/kg DM? Is the unit right?

Ln 199-200: Kirchgessner et al. is not listed with number 60 in the references. Please adjust.

Ln 226: McGeough et al. ist not number 60 in the references. Please adjust.

Ln 289: Kittelmann et al. ist not number 69 in the reference list. Please adjust. Please check the whole reference list.

Ln 272 - 425: You do not describe the opportunities for feeding feed additives and the impact of reducing emissions. What`s you opinion about that and why do you not mention that topic? 

It would be helpful for the reader if you could add a part at the ende of the manuscript where you could summarize the relevant methods and opportunities to reduce the methane emissions for practical farmers and a description of the reduction potential of the different opportunities.

Author Response

Reviewer 2

The manuscript "Enteric methane emissions and animal performance in dairy and beef cattle production: enumerating  the opportunities and impact of reducing emissions" is an interesting paper about methane production from livestock production and provide an overview about the biochemical process in the rumen and possible impact for a reduction in methane production. 

The title is very similiar to the paper from Knaap et al. 2014: Enteric methane in dairy cattle production: quantifying opportunities and impact of reducing emissions. Was that desired from you?

Thank you for your suggestion.

Changed the title “Enteric methane emissions and animal performance in dairy and beef cattle production: strategies, opportunities, and impact of reducing emissions”

Overall comment: In your title you would like to enumerate the opportunities and impact of reducing emissions but that is not very clear described in the manuscript. Please add one part where AU describe relevant methods and out of an practical point of view of reducing methane emissions from dairy and beef cattle.

Changed the title “Enteric methane emissions and animal performance in dairy and beef cattle production: strategies, opportunities, and impact of reducing emissions”

Ln 85: You mentioned that in a database a lot of studies examined the effects of mitigation strategies on enteric CH4 emissions..... Could you please cite the studies which were used. Added two references [16, 20]

110: Is it true that 1033 studies are available? Deleted a sentence.

Ln 131-133: You only describe the CH4-measurement methods RC, SF6 and GF but you do not describe the methane measurement with a laser detector. Why do you not describe or mention that measurement method because there are some paper about that measurement method in the literature available.

It is a great question. However, I couldn’t find the meta-analysis data compared to RC, SF6, GF, and laser detectors.

Ln 139: What do you mean by animal intake? Do you mean feed intake? Please change. Changed to animal feed intake.

Ln 148: CH4 has an energy content of 55.65 MJ/kg DM? Is the unit right?

Changed to methane has an energy content of 55.65 MJ/kg

Ln 199-200: Kirchgessner et al. is not listed with number 60 in the references. Please adjust. Changed the reference number.

Ln 226: McGeough et al. ist not number 60 in the references. Please adjust. Changed the reference number.

Ln 289: Kittelmann et al. ist not number 69 in the reference list. Please adjust. Please check the whole reference list. Changed the reference number.

Ln 272 - 425: You do not describe the opportunities for feeding feed additives and the impact of reducing emissions. What`s you opinion about that and why do you not mention that topic? 

It is a good question. However, feed additives itself has a wide range of data and it has already been published various scientists:

  • Feng, 2020 (Net reductions in greenhouse gas emissions from feed additive use in California dairy cattle).
  • Honan et al., 2021. Feed additives as a strategic approach to reduce enteric methane production in cattle: modes of action, effectiveness and safety. Animal production science.

It would be helpful for the reader if you could add a part at the ende of the manuscript where you could summarize the relevant methods and opportunities to reduce the methane emissions for practical farmers and a description of the reduction potential of the different opportunities.

Added a sentence. Many of the approaches discussed are only partially strategies, all approaches to reducing enteric CH4 emissions should consider the economic impacts on farm profitability and the relationships between enteric CH4 and other GHG. A numerous dietary mitigation interventions have been identified that could help reduce CH4 emissions, and other strategies presently being explored and identified. The greatest declines in CH4 emissions are likely to be achieved through a combination of approaches, including dietary modification and improved rumen fermentation for improving feed conversion efficiency.

Reviewer 3 Report

The review Enteric methane emissions and animal performance in dairy and beef cattle production: enumerating the opportunities and impact of reducing emissions presents a very much needed view on this topic.

Methane mitigation is a very important research topic in livestock, especially ruminant nutrition and production. 

As the authors state, much of the literature presents methane mitigation results focused on absolute amounts and a lower amount of reports focus on methane mitigation effects presented in relation to productivity (milk yield, weight gain, etc).

Therefore the present approach is highly appreciated by this reviewer as I have felt for a long time that this type of review was very much needed.

My only suggestion would be asking the authors to emphasize the need to present results of methane mitigation strategies in terms of animal productivity and not absolute methane yield. This should be clearly stated in summary and conclusion. Maybe adding a final paragraph before conclusion related to the pros and cons of presenting absolute values or values relative to animal production.

Other than that, I do not have further comments or suggestion.
It was a very readable and enjoyable lecture.

Great job!!

Author Response

Added a sentence: New technologies offer the potential to manipulate the rumen microbiome through genetic selection and varying degrees by various dietary intervention strategies to reduce CH4 emissions. Strategies to reduce GHG emissions, however, still need to be developed which increase ruminant production efficiency whereas reducing the production of CH4 from cattle, sheep, and goats.

Round 2

Reviewer 1 Report

I have no further comments.